# Measuring the Measures: Discriminative Capacity of Representational Similarity Metrics Across Model Families

**Jialin Wu[1], Shreya Saha[2], Yiqing Bo[1], Meenakshi Khosla[1,3]**

[1]Department of Computer Science and Engineering, UC San Diego
[2]Department of Electrical and Computer Engineering, UC San Diego
[3]Department of Cognitive Science, UC San Diego
San Diego, CA 92037
{jlwu,ybo}@ucsd.edu, ssaha@ucsd.edu, mkhosla@ucsd.edu

## Abstract

Representational similarity metrics are fundamental tools in neuroscience and AI, yet we lack systematic comparisons of their discriminative power across model families. We introduce a quantitative framework to evaluate representational similarity measures based on their ability to separate model families—across architectures (CNNs, Vision Transformers, Swin Transformers, ConvNeXt) and training regimes (supervised vs. self-supervised). Using three complementary separability measures—d-prime from signal detection theory, silhouette coefficients and ROC-AUC—we systematically assess the discriminative capacity of commonly used metrics including RSA, linear predictivity, Procrustes, and soft matching. We show that separability systematically increases as metrics impose more stringent alignment constraints. Among mapping-based approaches, soft-matching achieves the highest separability, followed by Procrustes alignment and linear predictivity. Non-fitting methods such as RSA also yield strong separability across families. These results provide the first systematic comparison of similarity metrics through a separability lens, clarifying their relative sensitivity and guiding metric choice for large-scale model and brain comparisons.

## 1 Introduction

Representational similarity metrics have become fundamental tools for understanding deep neural networks, enabling comparisons across architectures, layers, and even between artificial and biological vision systems. The field has developed diverse approaches to quantify representational similarity, including methods that utilize stimulus-by-stimulus similarity matrices to compare across networks, like Representational Similarity Analysis (RSA) as well as methods that learn explicit mappings to align neural dimensions, such as linear predictivity, Procrustes alignment [1, 2], and soft matching [3]. While these metrics have advanced our understanding of representational alignment, a critical question remains unexplored: how well do these metrics discriminate between different model families?

The current landscape of representation analysis is hindered by a fragmented collection of over 100 comparison methods [4, 5], many producing inconclusive results. This methodological chaos has real consequences: emerging evidence suggests that architectural differences between CNNs and transformers show negligible effects on brain alignment when assessed using prevailing metrics [6], raising the troubling possibility that our analytical tools lack the sensitivity to detect meaningful model differences. The field has focused intensely on benchmarking models against brain data using

single metrics, yet we lack systematic benchmarks of the metrics themselves. This gap is particularly concerning as researchers often default to popular choices without understanding what aspects of representations each metric captures or how sensitive they are to different model properties.

In this work, we introduce a systematic framework to evaluate the discriminative capacity of representational similarity metrics. We analyze 35 vision models spanning four architectural families (CNNs, Vision Transformers, Swin Transformers, ConvNeXt) and two training paradigms (supervised and self-supervised), using four similarity metrics. For each metric, we quantify its ability to separate model families using complementary measures from signal detection theory (d-prime), clustering quality (silhouette coefficients) and ROC-AUC.

## 2   Methods

We evaluate representational similarity metrics using 35 vision models spanning diverse architectural families and training paradigms. Our framework quantifies each metric's ability to discriminate between model families through complementary separability measures.

### 2.1   Model Selection and Dataset

We analyze 35 models across four primary categories: supervised CNNs, self-supervised CNNs, supervised Transformers, and self-supervised Transformers. We treat ConvNeXt [7] and Swin [8] as distinct families due to their hybrid nature—ConvNeXt incorporates Transformer-inspired design principles within a convolutional architecture, while Swin introduces CNN-like inductive biases into the Transformer framework. For evaluation, we use a curated subset of ImageNet-1k [9]. Complete model specifications and dataset details are provided in Appendices A and B.

### 2.2   Metrics for Representational Similarity

We evaluate widely used similarity metrics that differ in the flexibility of the mappings they permit—from rigid geometric transformations (Procrustes), to looser linear mappings (linear predictivity), to fully permutation-based alignments (soft-matching), as well as non-fitting approaches that compare representational geometry directly (RSA). Consider two representations $\mathbf{X}_i \in \mathbb{R}^{M \times N_i}$ and $\mathbf{X}_j \in \mathbb{R}^{M \times N_j}$ from different models, where $M$ denotes the number of stimuli and $N_i, N_j$ denote the number of units. All representations are mean-centered along the sample dimension as required.

**Representational Similarity Analysis (RSA) [10].**   RSA compares the geometry of representations by correlating their Representational Dissimilarity Matrices (RDMs). For each representation, we compute pairwise dissimilarities among all stimuli, creating an $M \times M$ RDM that captures the relational structure. The similarity between models is the Spearman correlation between their RDMs. This approach is invariant to orthogonal transformations and captures how models organize their representation spaces, independent of the specific features they use.

**Soft Matching [3].**   Soft Matching (SoftMatch) generalizes permutation distance [11] to representations with different numbers of units by relaxing permutations to "soft permutations." Specifically, The method seeks a mapping matrix $\mathbf{P} \in \mathcal{T}(N_i, N_j)$ in the transportation polytope [12], where each entry $P_{ij}$ represents the matching weight between units. The constraints ensure doubly-stochastic normalization: $\sum_{j=1}^{N_j} P_{ij} = \frac{1}{N_i}, \quad \sum_{i=1}^{N_i} P_{ij} = \frac{1}{N_j}$. The optimization problem is

$$d_T(\mathbf{X}_i, \mathbf{X}_j) = \min_{\mathbf{P} \in \mathcal{T}(N_i, N_j)} \sum_{k,l} \mathbf{P}_{kl} \| x_i^{(k)} - x_j^{(l)} \|^2,$$

where $x_i^{(k)}$ and $x_j^{(l)}$ are the $k$-th and $l$-th columns (units) of $\mathbf{X}_i$ and $\mathbf{X}_j$. The optimal transport plan $\mathbf{P}^\star$ is found via the network simplex algorithm. When $N_i = N_j$, this reduces to an optimal permutation. The final similarity score is the mean unit-wise correlation between $\mathbf{X}_j$ and $\mathbf{X}_i \mathbf{P}^\star$.

**Procrustes Alignment [1, 2].**   Procrustes analysis finds the optimal orthogonal transformation that aligns two representations while preserving their geometric structure. For representations with different dimensions, we first zero-pad the smaller representation. The method then seeks an orthogonal matrix $\mathbf{M} \in \mathcal{O}(N)$ that minimizes $\min_{\mathbf{M} \in \mathcal{O}(N)} \| \mathbf{X}_j - \mathbf{M} \mathbf{X}_i \|_2^2$ where $\mathcal{O}(N) = \{ \mathbf{M} \in \mathbb{R}^{N \times N} : \mathbf{M}^\top \mathbf{M} = \mathbf{I} \}$. This is solved via singular value decomposition, allowing rotations and

reflections while maintaining distances and angles. The alignment score is the correlation after optimal transformation.

**Linear Predictivity [13].** Linear predictivity imposes no constraints on the transformation, seeking any linear mapping $\mathbf{M} \in \mathbb{R}^{N_j \times N_i}$ that best predicts one representation from another: $\min_{\mathbf{M}} \|\mathbf{X}_j - \mathbf{M}\mathbf{X}_i\|_2^2$. Solved via ordinary least squares, this metric captures the maximum information overlap achievable through linear transformation, serving as an upper bound on representational similarity. Here again, we compute the final similarity as the Pearson correlation between the target representation and the optimally transformed source: $\text{corr}(\mathbf{X}_j, \mathbf{M}\mathbf{X}_i)$.

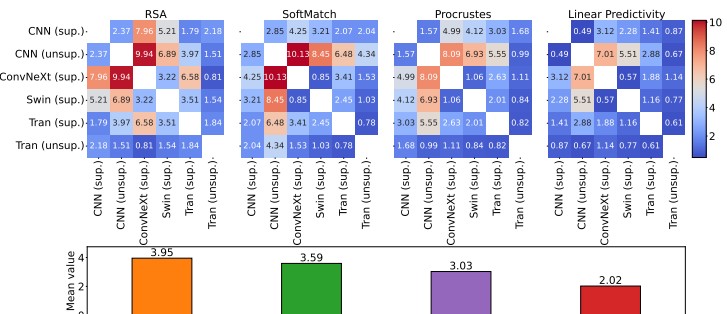

Figure 1: (Top) Heatmaps showing d-prime separability scores for pairwise comparisons between six model families: CNN (sup.), CNN (unsup.), ConvNeXt (sup.), Swin (sup.), Tran (sup.), and Tran (unsup.). Each panel corresponds to a different representational similarity metric: RSA, Soft Matching, Procrustes, and Linear Predictivity. Higher d-prime values (darker colors) indicate better separation between families, with d' > 2 conventionally considered strong discrimination. Each cell represents the averaged bidirectional d-prime between two model families. Diagonal entries are undefined (comparing a family with itself) and shown in white.

## 2.3 Metrics for Model Family Separation

We next describe the measures used to evaluate how well representational metrics capture separability between model families. Because family separation is inherently bidirectional, we compute directional scores (for d-prime and silhouette) in both directions and report the average as the final result.

**D-Prime (D') [13].** D' quantifies the separation between intra-family and inter-family similarity distributions. It is defined as $d' = \mu_{\text{within}} - \mu_{\text{between}} / \sqrt{0.5(\sigma_{\text{within}}^2 + \sigma_{\text{between}}^2)}$, where $\mu$ and $\sigma^2$ denote the mean and variance of the respective distributions. Higher values indicate tighter clustering within a family and greater spread across families, reflecting stronger separability.

**Silhouette Score [14].** For each model $i$, we compute the average distance $a(i)$ to all other models in the same family and the average distance $b(i)$ to models in the other family. The silhouette value is then $s(i) = (b(i) - a(i))/\max\{a(i), b(i)\}, s(i) \in [-1, 1]$. Values near 1 indicate that the model is well grouped with its own family, values near 0 suggest boundary placement, and negative values imply greater similarity to another family. The overall silhouette score is obtained by averaging $s(i)$ across all models.

**ROC-AUC [15].** ROC-AUC treats family separability as a binary classification problem between intra- and inter-family pairs. For each family, intra-family similarity scores are treated as positives and inter-family scores as negatives. The ROC curve summarizes the trade-off between true positive and false positive rates across thresholds, and the area under the curve (AUC) ranges from 0.5 (chance) to 1.0 (perfect separation). Unlike d-prime and silhouette, ROC-AUC is inherently symmetric, providing a robust global measure of discriminability.

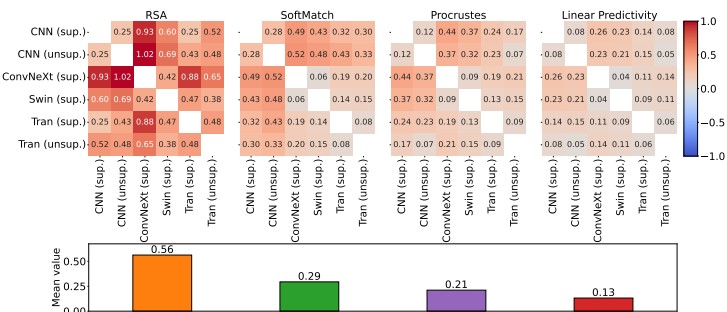

Figure 2: Same as Figure 1, but using silhouette scores instead of d-prime as the separability measure. Silhouette scores range from -1 to 1, where positive values (darker colors) indicate that models are well-clustered within their families and separated from other families, values near 0 suggest models lie at family boundaries, and negative values indicate misclassification.

## 3 Results

Our systematic evaluation reveals clear differences in the discriminative capacity of representational similarity metrics (Figures 1, 2). RSA demonstrates the strongest overall separability with a d-prime of 3.95, closely followed by SoftMatch ($d' = 3.59$), then Procrustes ($d' = 3.03$), with Linear Predictivity showing substantially weaker discrimination ($d' = 2.02$). The pattern is particularly pronounced for silhouette coefficients, where RSA achieves a notably high score of 0.56—indicating robust within-family clustering—while the mapping-based metrics show progressively weaker clustering: SoftMatch (0.29), Procrustes (0.21), and Linear Predictivity (0.13). ROC-AUC analysis confirms this hierarchy (Figure 3: RSA (0.9257) and SoftMatch (0.8994) achieve superior classification of within- versus between-family pairs, followed by Procrustes (0.8979) and Linear Predictivity (0.8137).

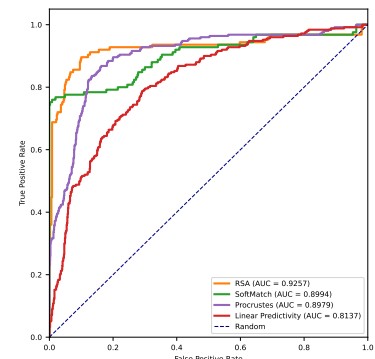

Figure 3: Global ROC curves comparing the discriminability of all representational similarity metrics. Each curve reflects the trade-off between true- and false-positive rates when distinguishing within-family from between-family pairs.

The hierarchy reveals an important principle: metrics that preserve representational geometry while allowing controlled flexibility achieve superior discrimination. RSA's strong performance across both measures suggests that comparing relational structure—how models organize their representation spaces—provides the most reliable signal for distinguishing model families. The mapping-based metrics show an inverse relationship between flexibility and discriminability: as we progress from the constrained SoftMatch through Procrustes to unconstrained Linear Predictivity, separability monotonically decreases. This finding challenges the assumption that looser metrics better capture representational differences. Instead, the constraints imposed by metrics like RSA and SoftMatch appear to filter out incidental variations while preserving the essential computational signatures that distinguish model families. Notably, even supervised and unsupervised variants within the same architecture family—traditionally considered highly similar—are reliably separated by RSA and SoftMatch, demonstrating the sensitivity of geometry-preserving metrics, or metrics sensitive to representational form, to architecture- or training-induced representational differences.

## 4 Discussion

Our findings highlight that representational similarity metrics differ in their ability to separate model families. Metrics imposing stronger alignment constraints provide higher discriminability than looser measures, and non-fitting approaches show strong separation. One limitation of our analysis is that

we are using human intuitions to impose a desideratum for evaluating metrics: that models within the same family should cluster together and separately from models from another family. However, sometimes representational convergence may emerge even between models from different families. Nevertheless, our findings still provide practical guidance for representation analysis: researchers should select metrics based on their discrimination goals rather than defaulting to popular choices. More broadly, our framework offers a principled way to benchmark similarity metrics and clarify their trade-offs, paving the way for more interpretable and goal-aligned comparisons across models and between models and brains.

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

# A   Experiment Settings

**Datasets.**   We use ImageNet-1K [9]. To control class imbalance and reduce compute, we evaluate on a balanced subset drawn from the test/validation split: 50 images per class sampled uniformly at random (fixed seed). All representational metrics are computed on this subset.

**Models.**   All models are trained on the ImageNet-1K training set. We obtain pretrained weights from `torchvision` [16], Torch Hub [17], `timm` [18], or the official repositories. Unless noted otherwise, we extract activations from each model's penultimate layer. For CNNs, which commonly include global average pooling, we use that pooled feature. For ViT-style models, we average non-CLS token embeddings to form the final representation for consistency across architectures.

# B   Model Family and Architecture Choices

We evaluate multiple architectures within each family to capture variation in depth, width, and design choices.

**Convolutional Neural Network (supervised; CNN (sup.)).**   Bottom-up hierarchies with convolutions and pooling that impose strong local inductive biases. We include AlexNet [19], VGG-11/13/16/19 (with/without batch normalization) [20], and ResNet-18/34/50/101/152 [21].

**Transformer (supervised; Trans (sup.)).**   Vision Transformers partition images into fixed-size patches and use multi-head self-attention for global interactions [22]. We include ViT-S/16, ViT-B/16, ViT-L/16, and ViT-B/32.

**ConvNeXt [7].**   A convolutional family inspired by Transformer design (e.g., large-kernel depthwise convolutions, patchified stems, inverted bottlenecks). We use ConvNeXt-Tiny/Small/Base/Large.

**Swin Transformer [8].**   A hierarchical Transformer with shifted window attention for efficient locality while retaining global context. We use Swin-Tiny/Small/Base/Large.

**Convolutional Neural Network (self-supervised; CNN (unsup.)).**   Methods trained without labels using CNN backbones (ResNet-50). We include MoCo [23], DINO [24], SwAV [25], and Barlow Twins [26], spanning contrastive and non-contrastive paradigms (momentum contrast, self-distillation, online clustering, redundancy reduction).

**Transformer (self-supervised; Trans (unsup.)).**   Label-free training with Transformer backbones. We include DINO-ViT-B/16 and DINO-ViT-S/16 [24], MoCo-ViT-B/16 [23], and MAE-ViT-B/16 and MAE-ViT-L/16 [27].

