# OpenReview forum: "Measuring the Measures: Discriminative Capacity of Representational Similarity Metrics Across Model Families"
_NeurIPS.cc/2025/Workshop/UniReps — UniReps2025_

### Official Review · Reviewer_fVjp · 2025-09-04
**How well do similarity measures cluster model families?**

**Confidence:** 4

**Review:**

**Summary:**


The work considers four representational similarity measures: RSA, Softmatch, Procrustes Alignment and Linear Predictivity.
They use these measures to compare representations from a number of different models grouped into six families. They use
three separability measures (d-prime, silhouette coefficients and ROC-AUC) to consider how well the similarity measures can
distinguish between these six model families.


**Strengths:**

**S1:** UniReps is about representational similarity. Notions of similarity depend on similarity measures.
	Thus, work on exploring similarity measures is extremely important.

**S2:** The chosen similarity measures have a nice spread of properties.

**S3:** The focus on separability of model families is interesting.



**Weaknesses:**

**W1:** It would be nice with a bit of discussion of what it means when a certain similarity measure can discriminate between
two model families and another cannot. For example, if linear predictivity finds two models similar, but RSA does not, then
the model representations can be transformed into each other via a linear transformation, but it is not an orthogonal transformation.
If RSA discriminates well between two model families, but linear transformation does not, then that is probably because the differences of
representations are often translations? But what does it mean if it is the other way around? For example in figure 1,
Linear Predictivity discriminates better between Tran (unsup.) and ConvNeXt (sup.) than RSA, what does that mean for the representations?

**A suggestion**

Sometimes training the exact same model using a different random seed for weight initialization will result in different representations.
It would be interesting to see how well various similarity measures discriminate between single model types. That is how well do various
measures cluster models which only vary by random seed.

**Score:**

4

**Topic Fit:**

3

---

### Official Review · Reviewer_T25q · 2025-09-09
**Highly relevant submission with good potential, could be made more solid**

**Confidence:** 4

**Review:**

**Quality:** The submission has several thorough experimental comparisons between representational similarity metrics. It is well-written and the figures are neat. However, I think some of the results could be made more rigorous through additional experimentation and by attempting to understand or theoretically explain the reasons behind differences between metrics.

**Clarity:** The writing is clear and all experimental details are provided. The descriptions and goals of the comparisons are clear.

**Originality:** To my knowledge, while several works have attempted to come up with a taxonomy of different similarity metrics [1], theoretical results on similarity [2,3], or overall goals of such research [4], there are not many consolidated, thorough comparisons of these metrics in a practical setting. Thus, I think this line of work is fairly original and timely. There are a couple of very relevant works that the authors could refer to for further experiments, in particular through a standard benchmark [5] and additional metrics or axes to compare metrics [6].

**Significance:** The work could be very impactful, particularly to the UniReps/representational similarity community, and help them make principled choices of metrics to use in their experiments.

**Strengths:**
* The writing is largely clear and the experimental pipeline is straightforward and comprehensive, with popular representational similarity metrics compared through several discriminability metrics.
* The findings are very relevant to neuroscience and interpretability researchers and clearly fit the theme of the workshop.
* The proposed comparison framework can be easily extended to include several other metrics, tasks, and model/networks, as the comparisons don't depend on any output task variables (although that might be a useful additional axis to consider, i.e., output similarity but functional dissimilarity).

**Weaknesses & Suggestions:**
* Some other standard metrics like CCA and CKA could be tested. The authors could consider referring to [5,6] for a benchmark they could use in addition to their current experiments and inspiration on additional comparisons respectively.
* From the paper, it seemed to me that only single instances of trained networks are compared, and intra-network/model variability has not been taken into account. This would be very important in assessing the robustness of a metric and also in understanding whether several instances of a network with the same architecture or learning rule actually have geometrically similar representations (the methods considered here mostly deal with geometry, see my point below about other metrics).
* While there are several experiments and clear findings, there is very little in terms of interpreting why these metrics differ in the way they do, or why specific metrics do well on certain comparisons. I think the paper could be made significantly more solid with a thorough understanding, ideally with theoretical justification, of why some metrics here are "better" or more discriminable than others. The authors have already started to provide some intuition (through their interpretation that metrics with stronger assumptions/more restrictions show better discriminability + that RSA does well across the board because of comparing relational structure).
* Most/all metrics considered here focus solely on geometric similarity of static representations. However, a key aspect of computation is "temporal" processing, and arguably, similarity metrics should focus on how representations evolve over time or through several layers to truly compare functional similarity (see also [7] for another perspective). Metrics such as Dynamical Similarity Analysis (DSA) [8] could be included in this framework, and the authors could also attempt to compare an aggregation of cross-layer representations using the existing metrics to capture this in some way as well.
* The models compared are restricted to one particular task and dataset, although as mentioned in the strengths, the framework can be easily extended to include more models.
* Only model-to-model comparisons have been explored so far, and keeping the overall goals of representational alignment [4] in mind, it would be nice to see some model-to-brain comparisons integrated in this framework as well (e.g., see [9] where they compare behaviourally similar networks trained with different learning rules to each other and also to neural data using geometric (CCA) and dynamic (DSA) measures).
* A key flaw, as alluded to in a previous comment, is that representations are taken from just the penultimate layer. Another interesting analysis could be to compare representational similarity layer-wise for all models using various metrics and understand both within-model variability (using a fixed baseline comparison) and also study cross-model cross-layer variability.

**Typos:**
* Line 24: "Whie" -> "While"
* Line 139: "seperate" -> "separate"
* Line 237: stray quotation marks at the end

**References:**
1. Cloos, Nathan, Guangyu Robert Yang, and Christopher J. Cueva. "A framework for standardizing similarity measures in a rapidly evolving field." arXiv preprint arXiv:2409.18333 (2024).
2. Williams, Alex H. "Equivalence between representational similarity analysis, centered kernel alignment, and canonical correlations analysis." bioRxiv (2024): 2024-10.
3. Harvey, Sarah E., David Lipshutz, and Alex H. Williams. "What representational similarity measures imply about decodable information." arXiv preprint arXiv:2411.08197 (2024).
4. Sucholutsky, Ilia, et al. "Getting aligned on representational alignment." arXiv preprint arXiv:2310.13018 (2023).
5. Bo, Yiqing, et al. "Evaluating representational similarity measures from the lens of functional correspondence." arXiv preprint arXiv:2411.14633 (2024).
6. Klabunde, Max, et al. "Resi: A comprehensive benchmark for representational similarity measures." arXiv preprint arXiv:2408.00531 (2024).
7. Hayne, Lucas, Heejung Jung, and R. Carter. "Does Representation Similarity Capture Function Similarity?." Transactions on Machine Learning Research (2024).
8. Ostrow, Mitchell, et al. "Beyond geometry: Comparing the temporal structure of computation in neural circuits with dynamical similarity analysis." Advances in Neural Information Processing Systems 36 (2023): 33824-33837.
9. Codol, Olivier, et al. "Brain-like neural dynamics for behavioral control develop through reinforcement learning." bioRxiv (2024): 2024-10.

**Score:**

3

**Topic Fit:**

3

---

### Official Review · Reviewer_aRoN · 2025-09-11
**Interesting and important evaluation of representational similarity metrics**

**Confidence:** 3

**Review:**

The paper evaluates four representation similarity metrics (RSA, Soft Matching, Procrustes Alignment, and Linear Predictivity) with respect to how well they can separate distinct model families.

Strengths:
- The paper is easy to read and clearly written.
- The topic is interested and very important - with the increasing number of different metrics, studies evaluating them and showing strengths and weaknesses of each of them are crucial.
- I think the results are very interesting, even though I have some questions/problems about the methodological design (see below).


Weaknesses (or more like unclarities and open questions):
- I am actually not convinced why the underlying assumption should necessarily be true - why should there be differences among model families. How do we know that the model family should be a key factor determining the similarities, and not others, for example model size, data used for training etc.?
- Why only supervised vs. unsupervised methods? There are many different unsupervised methods that could lead to very distinct representations.
- Does it make sense to average over all the categories? Shouldn't supervised vs. unsupervised be compared separately and the same for different model families? Aren't we loosing some information?

Line 132-133: "...supervised and unsupervised variants within the same architecture family—traditionally considered highly similar..." Do you have any reference for this?

**Score:**

3

**Topic Fit:**

3

---

### Official Review · Reviewer_m4u8 · 2025-09-14
**Review for the paper "Measuring the Measures: Discriminative Capacity of Representational Similarity Metrics Across Model Families"**

**Confidence:** 4

**Review:**

# Summary

This paper conducts a benchmarking of different representational similarity metrics, evaluating how good they are at separating model families.

# Strengths

1. The writing is clear with nice illustrations and mathematical formulae.

2. The theme is novel and relevant: it presents a comparison of the metrics used previously to compare model architectures, which would be uninformative if the metrics themselves cannot separate the models well.

3. The conclusions are sound and valuable, sparking further discussion: RSA is best, and there is an inverse trend between flexibility and separability, which is not expected given previous findings.

# Weaknesses

1. It is not stated clearly in the paper whether multiple copies of models with different random seeds are used. If not (i.e., only one instance per model), the results may not be statistically robust.

2. Related work on this topic, i.e., how other research tackles the problem of metric benchmarking, is lacking and would be more informative to the reader.

# Recommendation

I recommend acceptance of the paper to the workshop.

**Score:**

4

**Topic Fit:**

3